# Five-Year Mortality of Patients with Perioperative Myocardial Infarction After On-Pump Isolated or Combined Coronary Artery Bypass Graft Surgery: A Retrospective Propensity Score-Weighted Analysis

**DOI:** 10.3390/jcm14227970

**Published:** 2025-11-10

**Authors:** Christophe Beyls, Pierre Huette, Paul Luang, Hélène Kowalik, Chloé Andriamifidy-Berti, Mathieu Guilbart, Mickael Bernasinski, Patricia Besserve, Gilles Touati, Thierry Caus, Hervé Dupont, Yazine Mahjoub, Osama Abou-Arab

**Affiliations:** 1Department of Anesthesiology and Critical Care Medicine, Amiens University Hospital, F-80054 Amiens, France; 2UR UPJV 7518 SSPC (Simplification of Care of Complex Surgical Patients) Research Unit, University of Picardie Jules Verne, F-80000 Amiens, France; 3Department of Cardiac Surgery, Amiens University Hospital, F-80054 Amiens, France; 4MP3CV Laboratory, University Health Research Center, University of Picardie Jules Verne, F-80054 Amiens, France

**Keywords:** CABG, cardiac surgery, pMI, myocardial infarction, mortality, on-pump

## Abstract

**Background:** Coronary artery bypass grafting (CABG) is a widely used procedure to treat coronary artery disease, performed either alone or in conjunction with other cardiac procedures. Perioperative myocardial infarction (pMI) remains a major complication after on-pump CABG and negatively influences survival. Its reported incidence varies with the applied definition, and little is known about its prognostic effect in combined surgeries. **Objective:** The aim of this study was to investigate the impact of pMI on 5-year survival after isolated or combined on-pump CABG. **Methods:** We retrospectively reviewed adult patients undergoing urgent or elective CABG with cardiopulmonary bypass at Amiens University Hospital between 2013 and 2017. Diagnosis of pMI followed the Fourth Universal Definition of myocardial infarction. The primary outcome was all-cause death within 5 years. Logistic regression and Cox proportional hazards analyses were performed, and inverse probability weighting based on propensity scores was used to minimize confounding. **Results:** Out of 712 patients, 112 (16%) experienced pMI. Five-year mortality was 32% in the pMI subgroup compared with 11% in those without pMI (*p* < 0.001). Before adjustment, pMI was associated with higher long-term mortality (HR = 2.62, 95%CI [1.73–4.00], *p* < 0.001). This effect persisted after weighting (HR = 2.43, 95% CI [1.56–3.78], *p* = 0.041). A landmark analysis excluding the first 30 postoperative days showed no significant link with later mortality (HR = 1.13, 95%CI [0.54–2.34], *p* = 0.74). Independent predictors of pMI included active smoking (OR = 2.24, 95% CI [1.36–3.69], *p* = 0.001) and prolonged bypass duration (>180 min) (OR = 2.57, 95% CI [1.19–5.34], *p* = 0.015). **Conclusions:** When defined by the Fourth Universal Definition, pMI was independently related to increased 5-year mortality following CABG, explained mainly by deaths occurring early after surgery.

## 1. Introduction

Coronary artery bypass grafting (CABG) is a common cardiac procedure in Europe, known for its low in-hospital mortality rate despite the increasing risk profile of patients [1]. It is the preferred treatment for multivessel coronary artery disease, whether performed as a standalone procedure or in conjunction with other cardiac interventions [2,3]. Diagnosing perioperative myocardial infarction (pMI) is a critical concern in CABG due to its association with increased mortality [4]. pMI is primarily identified through elevated cardiac biomarker levels. While creatine kinase MB (CK-MB) was historically the diagnostic standard, contemporary practice favors cardiac troponin [5]. However, the appropriate threshold for cardiac troponin levels (ranging from 10 to 70 times the upper reference limit) to diagnose pMI remains contentious [6].

According to the Fourth Universal Definition of Myocardial Infarction (4UD), pMI is diagnosed when cardiac troponin levels exceed ten times the upper reference limit, accompanied by evidence of new myocardial ischemia observed through electrocardiographic (ECG) and transthoracic echocardiography (TTE) within the initial 48 h after CABG [5]. This nuanced criterion adds complexity to the ongoing debate surrounding pMI diagnosis.

As a result of these complexities, reported pMI incidence in clinical trials varies significantly, ranging from 0.7% to 11.8% in CABG combined with valvular surgery [4]. This variability highlights the inherent challenges associated with pMI diagnosis, arising from inconsistencies in troponin threshold selection, numerous definitions, and irregular data collection post-cardiac surgery [7]. Such variability could lead to the underreporting of pMI cases, hindering our comprehensive understanding of its clinical implications and its association with perioperative mortality.

Recent studies have indicated that pMI, defined by the 4UD, is associated with increased 5-year all-cause mortality in isolated CABG cases [6]. However, data on this topic, particularly in combined cardiac surgery, are limited.

In this study, we aim to explore the relationship between pMI, defined by the 4UD criteria, and 5-year all-cause mortality in a large retrospective cohort of real-world patients undergoing isolated or combined CABG. We hypothesize that patients who experience pMI will have higher 5-year all-cause mortality rates, shedding light on the long-term consequences of this perioperative complication.

## 2. Methods

### 2.1. Study Population and Ethics

Adult patients undergoing elective or urgent on-pump CABG, either combined with valvular and/or ascending aortic surgery or performed as isolated procedures, were included in the study. Patients lacking sufficient data (TTE and ECG) for pMI diagnosis and those with missing records during follow-up were excluded.

The urgency of surgery was defined according to international guidelines. Elective procedures referred to stable patients scheduled in advance without ongoing ischemia or hemodynamic compromise. Urgent surgeries were performed during the same hospitalization in patients with ongoing or recurrent ischemia, hemodynamic instability, or failed percutaneous or medical therapy requiring surgical revascularization within 24 h [8].

### 2.2. Ethics

This is a retrospective study of cardiac surgery patients hospitalized in the CTVR-ICU at Amiens University Hospital during 2013–2017. The study was conducted by the Declaration of Helsinki and approved by the Institutional Review Board of Amiens University Hospital (Number: PI2022-843-0071) for non-interventional studies involving humans. Informed consent was waived by French law on clinical research for non-interventional studies [9]. Reporting is according to a guideline on reports of propensity score analysis [10] and STROBE statement of retrospective study [11].

### 2.3. pMI Definition

pMI was defined according to the 4UD from the European Society of Cardiology [5]. It required a cardiac troponin (cTnI) elevation of more than ten times the 99th percentile, along with at least one of the following criteria within 48 h of the primary cardiac surgery: (1) appearance of a new Q wave on ECG, (2) de novo occlusion postoperatively confirmed by coronary angiography, or (3) evidence of new abnormality in segmental contractility or loss of viable myocardium, consistent with an ischemic origin, observed on echocardiography.

At our center, cTnI levels were measured using conventional chemiluminescence immunoassay (ELISA Sandwich on Atellica^®^ Immunoassay analyzer, Siemens, Erlangen, Germany) at 0, 6, 24, and 48 h after the primary cardiac surgery. The 99th percentile threshold was set at 0.045 g/L. No sex-specific reference limits were applied, as a single institutional cut-off value was used. ECGs were performed at 0, 6, 24, and 48 h, and segmental kinetic abnormalities were assessed through TTE at 0, 24, and 48 h.

For emergency coronary surgeries performed in the context of an ongoing acute coronary syndrome, postoperative cTnI levels were not used for the diagnosis of perioperative myocardial infarction, as preoperative ischemia or infarction typically leads to pre-existing troponin elevation. In such cases, only new ischemic ECG changes and/or new regional wall motion abnormalities on echocardiography were considered for the diagnosis of pMI, in accordance with current recommendations [5].

### 2.4. Outcomes

The primary outcome was five-year all-cause mortality, defined as all-cause death after cardiac surgery. The secondary outcome was prognostic factors of the occurrence of pMI.

### 2.5. Data Collection

Clinical, biological, echocardiographic, and operative data were collected prospectively in a computerized database. Preoperative data collected were age, sex, weight and height, history of MI, CABG surgery or coronary angioplasty, and cardiovascular risk factors (atrial fibrillation, diabetes, dyslipidemia, hypertension, active smoking, previous stroke, chronic obstructive pulmonary disease, chronic renal failure), the presence of preoperative dyspnea according to the NYHA scale and angina symptomatology classified according to the Canadian classification [3]. The EuroSCORE II was calculated from the preoperative clinical data [12]. Intraoperatively, the type of surgery performed (presence of a valve, aortic tube, number of grafts, type of grafts), the duration of the pump, the time of aortic clamping, the administration of blood products, and the placement of invasive mechanical support (intra-aortic counterpulsation balloon and an extracorporeal oxygenation membrane) were collected.

In the postoperative period, the occurrence of a postoperative complication defined by the European Society of Anaesthesia and Resuscitation recommendations was collected [13]. According to guidelines, acute kidney injury (AKI) was defined according to KDIGO guidelines at least an increase in creatinine within 48 h over 26.5 µmol L^−1^. In-hospital mortality was defined as any postoperative death occurring within the hospital stay of the primary cardiac surgery.

### 2.6. Follow-Up

Follow-up was conducted using information from our institution’s medical records, input from referring cardiologists, phone conversations with patients or their family members, and data from the French National Institute of Statistics and Economic Studies database (INSEE). Notably, patients were kept from follow-up, ensuring a comprehensive follow-up rate of 100%.

### 2.7. Statistical Analysis

As appropriate, categorical variables are expressed as frequencies and percentages and compared using the χ2 or Fisher exact test. Continuous variables are expressed as mean± standard deviation (SD) and compared by unpaired Student *t*-test or the Mann–Whitney test. The patients were divided into two groups: the pMI group and the no-pMI group.

### 2.8. Missing Data and Propensity Score

Missing data were imputed for the following stages using predictive mean matching imputation. Five imputation data sets were produced. Missing data were analyzed as missing-not-at-random according to the Rubin rule [14]. Prognostic variables related to five-year all-cause mortality at 20% in univariate analysis were included in the propensity score (regardless of their differences between the two groups): Euroscore II, age, medical history of chronic renal failure and acute heart failure, clinical symptoms and type of MI before surgery, preoperative estimated glomerular filtration rate, preoperative hemoglobin, red blood cell transfusion and redo surgery for bleeding.

The probability of developing 5-year all-cause mortality for each patient has been estimated using logistic regression. For the average treatment effect on the entire population analysis, weights were attributed to each patient of the pMI and no pMI groups, making the two groups similar for the variables in the propensity score. These weights were calculated using the stabilized inverse probability of treatment weighting (IPTW). Balance (standardized mean differences lower than 10%) was checked for each variable [15]. HRs were estimated before and after weighting on the propensity score using Cox proportional hazards analysis. The Kaplan–Meier method plotted the survival curves at five years between the two groups, which were compared with the log-rank test.

In the pMI group, excess mortality was primarily concentrated in the immediate postoperative phase. To better isolate the impact of pMI on 5-year all-cause mortality among patients who survived the critical perioperative period, a landmark analysis was conducted starting at 30 days post-surgery. This approach excludes early in-hospital deaths and provides a focused evaluation of the long-term prognostic effect of pMI among survivors. Additionally, we analyzed the association between 5-year all-cause mortality and patients who underwent CABG alone

### 2.9. pMI Factors

Multivariate logistic regression methods were used to determine the predictors of pMI, which are expressed as odds ratios (OR) and 95% confidence intervals (CI). We dichotomized continuous variables based on thresholds reported in the literature as associated with increased mortality, including EuroSCORE 2 > 4% (intermediate and high-risk patients) [12], hemoglobin < 10 g/dL [16], CPB time > 180 min [17], and cross-clamping time > 60 min [18]. Model discrimination was assessed by the area under the receiver operating characteristic curve and by the Hosmer–Lemeshow goodness-of-fit statistic calibration. Perioperative factors with a univariate *p* < 0.05 were submitted to a logistic regression model by stepwise selection.

All tests were two-sided, and the threshold for statistical significance was set to *p*  <  0.05. Statistical analysis was performed with R studio software for macOS (version 1 September 2021 +  372) and its «dplyr», «ggplot2», «survminer», «survival», «hrbrthemes», «tableone», «ggeffects», «WeightIt», «cobalt», «compareGroups», «mice» and «epiR» and “reshape2” packages.

## 3. Results

From January 2013 to December 2017, 847 consecutive patients were hospitalized for on-pump isolated or combined CABG. Emergency and scheduled cardiac surgery were included. Among the 820 patients who met the inclusion criteria, 712 patients were analyzed. One hundred twelve patients (16%) experienced pMI according to the 4UD (pMI group), and 600 (84%) patients experienced no pMI (no-pMI group) (see Figure 1).

### 3.1. Patients Characteristic

Demographic and preoperative data are summarized in Table 1. In the pMI group, patients presented angina symptoms with a CCS class > II (*n* = 57/112 vs. 304/600, *p* = 0.04) and active smoking (*n* = 83/112 vs. *n* = 94/600, *p* = 0.008) compared to patients in the no-pMI group. In the pMI group, the EuroSCORE II was higher than in the no-pMI group (respectively, 6.5 ± 4 % vs. 5.3 ± 3 %, *p* = 0.01)

### 3.2. Perioperative, Postoperative Data and Follow-Up 

Concerning the type of cardiac surgery, there was no significant difference between the procedures, either for isolated CABG (*n* = 425/600, 71% vs. *n* = 81/112, 72%; *p* = 0.91) or for combined surgeries (*n* = 175/600, 29% vs. *n* = 31/112, 28%; *p* = 0.82; see Table 2). In the pMI group, the cardiopulmonary bypass time was longer than in the no-pMI group (respectively, (104 ± 56 min vs. 89 ± 41 min, *p* = 0.04). Patients in the pMI group received more red blood cell products (*n* = 31/112, 28% vs. *n* = 103/600, 17%; *p* = 0.01) and vasopressors (*n* = 39/112, 35% vs. *n* = 129/600, 21%; *p* = 0.003) compared to those in the no-pMI group. In the pMI group, patients required mechanical support more frequently (8%, n = 9/112) compared to those in the no-pMI group (02%, *n* = 10/600; *p* < 0.0001). Patients with pMI experienced higher rates of AKI (*n* = 36/112, 32% vs. *n* = 97/600, 16%; *p* = 0.0001) and underwent more revision surgeries (*n* = 15/112, 14% vs. *n* = 28/600, 5%; *p* = 0.002) during their ICU stay. During the follow-up period (mean 5.6 ± 2.2 years), 102 patients died (15%), including 35% (*n* = 36/103) during their in-hospital stay (Table 2).

### 3.3. Association Between pMI and 5-Year All-Cause Mortality

After IPTW adjustment, the mean standardized difference for pre- and peri-operative variables associated with 5-year all-cause mortality was <10%, as shown in Appendix A, and illustrated in a Love plot (Figure 2A). Before weighting, the 5-year all-cause mortality rate was significantly different higher in the pMI group (32%, *n* = 36/112) than in the no-pMI group (11%, *n* = 66/600) (*p* = <0.0001). pMI was significantly associated with 5-year all-cause mortality with an HR of 2.62 95%CI = [1.73–4.00] (*p*<0.001). After weighting, the association remained significantly important, with an HR at 2.43 [1.56–3.78] (*p*  =  0.041). The time to 5-year survival was favored considerably in patients without pMI pre- and post-weighting (Figure 2B).

### 3.4. Isolated CABG

In the population undergoing isolated CABG (71%, *n* = 506/712), the 5-year all-cause mortality rate was notably higher in the pMI group (31%, *n* = 25/81) compared to the no-pMI group (12%, *n* = 48/425), with a distinct favoring of the 5-year survival curve (Appendix A) among patients without pMI (Log Rank test with *p* < 0.001). Combined procedures were not associated with the occurrence of pMI (Table 3).

### 3.5. Impact of pMI After 30 Days

Given the significant number of deaths during the immediate postoperative period, we performed a landmark analysis to exclude postoperative deaths and assess the impact of pMI on patients who survived hospital discharge. In the landmark analysis, 5-year mortality was similar between the two groups, with overlapping survival curves (log-rank test: *p* = 0.47, Appendix A). After weighting, the HR for the pMI group was 1.13 (95% CI: 0.54–2.34, *p* = 0.74), indicating no significant difference in long-term mortality between the groups.

### 3.6. Prognostic Factors of pMI

Multivariable logistic regression analysis identified active smoking (OR [95% CI] = 2.24 [1.36–3.69], *p* = 0.001) and a cardiopulmonary bypass time > 180 min (OR [95% CI] = 2.57 [1.19–5.34], *p* = 0.015) as the independent factors associated with pMI (Table 3).

## 4. Discussion

In this retrospective single-center cohort comprising 712 patients who underwent isolated or combined CABG, pMI defined by 4UD occurred in 16% of cases. Remarkably, pMI emerged as an independent predictor of 5-year mortality, a significant finding observed before and after weighting. However, this relationship is primarily driven by excess mortality during the immediate postoperative period, suggesting that long-term mortality is not significantly increased for patients discharged home, even if they experienced a pMI.

These findings emphasize the need to improve perioperative management to reduce the incidence of pMI, while indicating that, in this cohort, no significant long-term difference in survival was observed among hospital survivors.

### 4.1. Incidence of pMI

Our data emphasize the significant influence of pMI after CABG on patients’ long-term prognosis. While evidence exists regarding the impact of pMI after isolated CABG, knowledge about long-term outcomes for pMI after CABG in combined procedures is limited. The true frequency of pMI remains uncertain because the incidence of pMI following CABG shows substantial variation depending on the diagnostic criteria used, ranging from 0% to 29% [19]. The recent controversy surrounding the EXCEL trial underscores the notable disparities in pMI rates associated with the chosen definition and underscores the diagnostic challenges encountered even in multicenter randomized controlled trials [7].

In our study, we defined pMI using the 4UD and observed an incidence of 16%. Our findings closely paralleled those of Pretto et al., who reported a pMI incidence of 24.1% using the 3rd Universal Definition (3UD), which shares similar diagnostic criteria with the 4UD [5,20]. However, our observed incidence notably exceeded that reported by Litwinowicz et al., who identified a pMI rate of 3% within a retrospective cohort comprising 4642 patients undergoing isolated CABG [21]. It is noteworthy that their study cohort exhibited lower surgical risk, as reflected by lower EuroSCOREs, with means at 2.72 ± 1.21 in the pMI group and 2.18 ± 1 in the non-pMI group, in contrast to our study (6.5 ± 5 and 5.3 ± 3, respectively). Moreover, Litwinociz et al. specifically focused their research on isolated CABG cases [21]. Furthermore, it is pertinent to recognize a plausible factor contributing to the disparities in pMI incidence. In Litwinowciz’s study, the assessment of cardiac troponin levels occurred beyond the initial 12 h postoperative period, contingent solely upon clinical necessity [21]. This methodology potentially resulted in an underestimated representation of the actual pMI incidence.

### 4.2. pMI and 5-Year Mortality

In our study, we observed a five-year all-cause mortality rate of 15%, which was significantly higher in the pMI group. These results align with similar findings from research studies examining 5-year all-cause mortality after cardiac procedures and utilizing the 4UD criteria. For instance, the study from Carr et al. [22], encompassing 1028 patients whom underwent CABG, reported a 17% 5-year mortality rate. Similarly, in the study from Pölz et al., involving 2829 patients, a 16% 5-year mortality rate was documented [6]. In the pMI group, the 5-year all-cause mortality was 32%, with an adjusted HR of 2.43 [1.56–3.78], indicating a higher risk of death (*p*   =   0.041). For instance, in the Pölz study, the 5-year all-cause mortality rate was 60% among patients with pMI, with an HR of 2.13; 95% CI 1.19–3.81; *p* = 0.011 [6]. Similarly, in the Litwinowciz study, patients with pMI experienced a 56% mortality rate, associated with an HR of 1.942, 95% CI: 1.277–3.073, *p* = 0.005 [21].

Using a landmark analysis, we observed that the five-year mortality associated with pMI was primarily driven by an excess mortality in the immediate post-operative period. After 30 days, pMI was no longer significantly associated with five-year mortality, contrary to the findings of Carr et al. [22], who reported an increase in MACE events after 30 days in their landmark analysis. However, their analysis focused exclusively on MACE outcomes, including mortality, rather than all-cause mortality. Upon examining the Kaplan–Meier curves and considering the relatively low number of deaths in their cohort, it is likely that all-cause mortality differences after 30 days were not significant.

The reasons behind the elevated mortality observed in patients having pMI remain unclear. There are several potential explanations for this phenomenon. One significant issue in understanding pMI and providing specific care is that only some patients undergo coronary angiography to determine its cause. In our study, only 1% of patients had this procedure. This low rate may be because our hospital does not routinely use coronary angiography as the first choice for diagnosis, except in cases of acute thrombosis or strong signs of graft problems.

In contrast, some other hospitals use different approaches that lead to more frequent use of coronary angiography. In Rupprecht et al.’s study, coronary angiography frequently revealed frequent graft pathologies [23]. These findings underscore the potential benefit of establishing standardized clinical protocols and adopting advanced cardiac imaging methods, such as coronary computed tomography scans, to enhance our comprehension of pMI etiology and optimize treatment strategies. Moreover, patients with pMI might undergo incomplete revascularization, a well-known risk factor for post-CABG mortality, leading to a less favorable prognosis. Another hypothesis is that the occurrence of heart failure after pMI could negatively impact prognosis. However, a comprehensive understanding of this issue awaits further investigation in future trials to unravel the underlying mechanisms at play.

### 4.3. Factors of pMI

In our study, preoperative active smoking increased the risk of pMI (OR 1.97, 95% CI [1.12–3.19], *p* = 0.006). Smoking is a well-known independent risk factor for cardiovascular mortality and has been associated with an increased risk of pMI in the preoperative period [24]. This highlights the significance of smoking cessation, with a strong emphasis on preoperative intervention, in line with 2021 American College of Cardiology guidelines [25]. Prolonged cardiopulmonary bypass (>180 min) was also associated with pMI (OR 2.51 [1.19–5.25], *p* = 0.01). A 2012 study by Mejía et al. found a significant increased risk of perioperative myocardial infarction when cardiopulmonary bypass exceeded 108 min (OR 40, 95% CI [2.7–578], *p* = 0.007) in patients undergoing urgent coronary artery bypass grafting [26]. Longer cardiopulmonary bypass durations lead to heightened inflammation, systemic injury due to glycocalyx impairment [27], and an increased risk of vasoplegia shock leading to an increased risk of mortality.

### 4.4. Clinical Implications

From a practical standpoint, our results emphasize the importance of preoperative identification of patients at high risk of pMI, such as those with recent ACS, diffuse coronary disease, or prolonged anticipated clamp times. In these patients, adopting tailored surgical strategies—including minimizing ischemic time or using less invasive or off-pump techniques when feasible—may help reduce myocardial injury. At the institutional level, standardized perioperative pathways integrating systematic postoperative troponin surveillance and predefined diagnostic thresholds (Δ-troponin) could allow early detection of pMI. In line with recent EACTS recommendations, this should prompt early coronary angiography or coronary CT angiography to confirm graft patency and guide reintervention when necessary [28]. Such structured protocols could improve short-term outcomes by enabling prompt correction of reversible ischemia. In the longer term, implementing dedicated follow-up programs for patients with preoperative ACS or pMI—including multimodal imaging and aggressive secondary prevention—may mitigate the sustained risk of adverse events. These practical measures bridge the gap between research findings and real-world clinical application, providing a foundation for structured prevention and management of pMI after CABG.

## 5. Limitations

Our study has inherent limitations due to its retrospective nature and being conducted at a single center, which may introduce a higher risk of missing data. However, it is essential to note that in our center, the perioperative management of cardiac surgery is protocolled, ensuring comprehensive electronic collection of all clinical, biological, and hemodynamic data, particularly during the intensive care unit stay.

In this study, we examined the impact of pMI in a mixed population undergoing both isolated CABG and combined cardiac procedures, which could potentially introduce bias. However, combined procedures were evenly distributed between groups and were not associated with either pMI or 5-year all-cause mortality, suggesting that surgical heterogeneity did not materially influence our findings. In addition, in the era of percutaneous revascularization, the proportion of combined surgeries continues to rise, and no data on the impact of pMI were reported. For instance, in the United Kingdom, the proportion of combined surgeries was 12% (*n* = 3274/26,879) in 2002 and increased to 21% (*n* = 4113/19,191) by 2016 [29].

Approximately one-third of patients underwent CABG in the context of an acute coronary syndrome, predominantly NSTEMI. Although this variable was not independently associated with pMI in our cohort, this may be partly explained by the limited number of patients and the resulting lack of statistical power. This subgroup likely reflects increased myocardial susceptibility to perioperative ischemic injury, consistent with findings in NSTEMI patients undergoing PCI, where periprocedural myocardial injury independently predicted adverse outcomes [30].

There are other limitations to consider: we reported only all-cause mortality and not specific cardiovascular deaths or composite criteria of Major Adverse Cardiac and Cerebrovascular Events. This limitation is primarily due to the lack of available data in the INSEE database. Nevertheless, it is important to highlight that this constraint is encountered in several research studies addressing this topic [21].

In addition, although pMI was mainly associated with early mortality, the absence of a significant difference beyond 30 days should be interpreted with caution. The number of late events after 30 days was relatively small, which may have limited the statistical power of the landmark analysis and precluded detection of subtle differences in long-term mortality among survivors. Finally, the diagnosis of pMI, which involves ECG and echocardiographic assessments, integral components of the 4UD, may have been influenced by inter-operator variability. However, this potential bias is likely minimal, given the presence of highly experienced echocardiography specialists and cardiologists within our intensive care team.

## 6. Conclusions

Our research provides real-world evidence that post-CABG pMI, as defined by the 4UD criteria, significantly worsens long-term prognosis, primarily due to excess mortality in the immediate post-operative period, regardless of whether the procedure is performed in isolation or combined with other cardiac interventions. Among patients surviving this critical period, long-term outcomes appear comparable, suggesting that the prognostic impact of pMI is largely confined to the immediate postoperative window. There is an urgent need for further studies to clarify the underlying mechanisms of pMI and to develop tailored interventions aimed at optimizing post-operative outcomes.

## Figures and Tables

**Figure 1 jcm-14-07970-f001:**
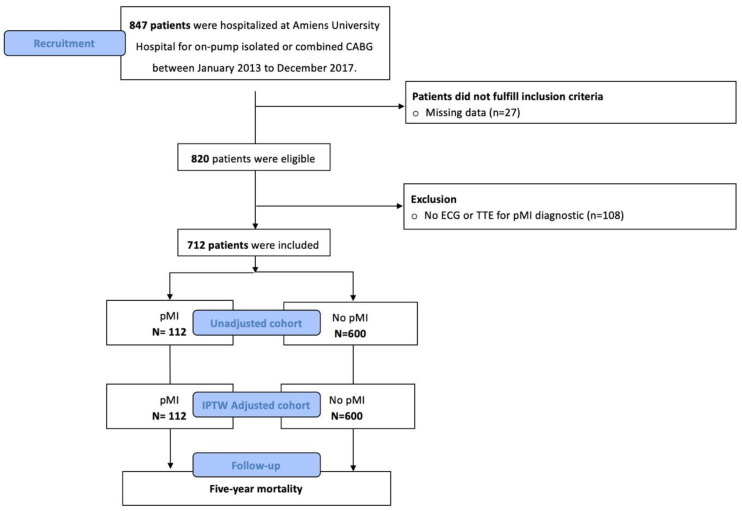
Flow chart of the study. IPTW: inverse probability of treatment weighting; pMI: perioperative myocardial infarction.

**Figure 2 jcm-14-07970-f002:**
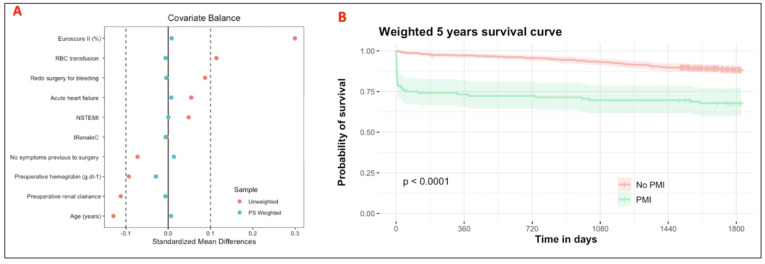
(**A**) Love plots for standardized mean differences comparing covariate values before (orange circle) and after (green circle) propensity score weighting for assessing 5-year mortality. (**B**) Propensity weighted Kaplan–Meier 5-year survival curve according to the occurrence of pMI.

**Table 1 jcm-14-07970-t001:** Demographic and Preoperative characteristics of the study population.

Variables	No pMI (*n* = 600)	pMI (*n* = 112)	*p* Value	SMD
Age (years)	68 ± 9	67 ± 11	0.29	0.129
BMI (kg.m^−2^)	29 ± 11	29 ± 5	0.28	0.024
Male gender (*n*; *%)*	488 (81)	85 (76)	0.22	0.133
**Medical history, *n* (%)**				
No symptoms	97 (16)	10 (9)	0.05	0.220
Angina severity according to CCS > 2	304 (51)	57 (51)	1.00	0.005
Recent myocardial infarction	140 (23)	16 (14)	0.71	0.038
Previous PCI	107 (18)	28 (17)	1.000	0.001
Peripheral vascular disease	87 (14)	19 (17)	0.59	0.068
Hypertension	477 (79)	83 (75)	0.21	0.128
Active Smoking	94 (16)	29 (27)	0.008	0.277
Diabetes mellitus	240 (40)	44 (39)	0.91	0.015
Dyslipidemia	446 (74)	78 (70)	0.29	0.105
Chronic renal disease	61 (10)	11 (10)	1.000	0.012
Stroke	36 (6)	6 (5)	1.000	0.042
Atrial fibrillation	81 (13)	16 (14)	0.89	0.018
Chronic obstructive pulmonary disease	59 (10)	6 (5)	0.15	0.170
Prior cardiac surgery	6 (1)	2 (2)	0.36	0.067
EuroSCORE 2 (%)	5.3 ± 3	6.5 ± 4	0.01	0.302
Hemoglobin (g/dL)	13.3 ± 1.8	13.1 ± 1.9	0.43	0.113
Glomerular filtration rate (mL/min/1.73 m^2^)	80 ± 27	72 ± 29	0.41	0.120
**Preoperative TTE**				
Left ventricular ejection fraction, (%)	56 ± 11	54 ± 13	0.19	0.183
**Acute coronary syndrome, *n* (%)**				
NSTEMI	121 (23)	28 (29)	0.24	0.116
STEMI	38 (7)	8 (8)	0.67	0.035
**Coronagraphy**				
One diseased vessel	58 (10)	1 (5)	0.20	0.127
Two diseased vessels	134 (22)	20 (18)	0.31	0.090
Three diseased vessels	405 (68)	84 (75)	0.14	0.164

Data are expressed in number, count (%) and mean (±deviation standard). BMI: body mass index, CABG: coronary artery bypass grafting CCS: Canadian Cardiovascular Society; NSTEMI: non-ST elevation myocardial infarction; PCI: percutaneous coronary intervention; SMD: mean standardized difference; STEMI: ST-elevation myocardial infarction; TTE: trans-thoracic echocardiography.

**Table 2 jcm-14-07970-t002:** Perioperative complications and outcomes.

Variables	No pMI (*n* = 600)	pMI (*n* = 112)	*p* Value	SMD
**Cardiac surgery procedure, *n* (%)**				
Isolated CABG	425 (71)	81 (72)	0.91	0.025
Complete arterial grafting	333 (55)	62 (55)	1	0.003
Right internal mammary artery	326 (54)	59 (53)	0.75	0.033
Left internal mammary artery	560 (93)	108 (96)	0.28	0.140
Radial-artery graft	33 (5)	6 (5)	1	0.006
Saphenous vein graft	251 (42)	47 (42)	0.83	0.021
Combined CABG, *n* (%)	175 (29)	31 (28)	0.82	0.152
Valve replacement	173/175 (98)	30/31 (97)	0.73	0.046
Aortic root replacement	25/175 (14)	3/31 (10)	0.61	0.082
**Number of grafts, *n* (%)**				
>3	336 (56)	61 (54)	0.68	0.040
CPB time, (min)	89 ± 41	104 ± 56	0.04	0.270
Cross aortic clamp, (min)	61 ± 34	64 ± 41	0.97	0.084
Administration of RBC products, *n* (%)	103 (17)	31 (28)	0.01	0.257
Administration of vasopressor, *n* (%)	129 (21)	39 (35)	0.003	0.395
Mechanical circulatory support, *n* (%)	10 (2)	9 (8)	<0.0001	0.302
ECMO	3 (30)	3 (33)	0.05	0.174
IABP	7 (70)	8 (88)	0.001	0.291
Cardiac troponin at day 2 (µg/L)	4.6 ± 10.8	20.5 ± 75.8	0.001	0.295
**Postoperative complications, *n* (%)**				
Atrial fibrillation	152 (25)	28 (25)	1	0.003
Stroke	9 (1)	3 (3)	0.41	0.084
Mechanical ventilation > 24 h	49 (8)	29 (26)	<0.0001	0.491
Pneumonia	92 (15)	21 (19)	0.32	0.099
Acute kidney injury	97 (16)	36 (32)	<0.0001	0.385
Vasoplegic syndrome	179 (30)	44 (39)	0.05	0.218
Revised surgery	28 (5)	15 (14)	0.002	0.307
**Outcomes**				
Intra-hospital death, *n* (%)	8 (1)	28 (25)	<0.0001	0.718
Five-year death, *n* (%)	66 (11)	36 (32)	<0.0001	0.509
Length of ICU, days	5 ± 20	5 ± 10	0.04	0.117
Length of hospital, days	13 ± 10	13 ± 13	0.08	0.021

Data are expressed in number (%) and mean (±standard deviation). CABG: coronary artery bypass grafting; CPB: cardiopulmonary bypass; ECMO: extracorporeal membrane oxygenation; IABP: intra-aortic balloon pump; ICU: intensive care unit; pMI: post-operative myocardial infarction; RBC: red blood cell; SMD: standardized mean difference.

**Table 3 jcm-14-07970-t003:** Factors associated with pMI in patients with isolated or combined CABG.

Variable	OR (95% CI) by Logistic Regression			
	Univariate Analysis		Multivariate Analysis	
	OR (95%CI)	*p*	OR (95%CI)	*p*
EuroSCORE II > 4%	1.51 (1.01–2.25)	0.048	1.45 (0.94–2.25)	0.08
Diabetes mellitus	0.93 (0.64–1.34)	0.71	-	
Dyslipidemia	0.79 (0.51–1.23)	0.32	-	
Active Smoking	1.97 (1.12–3.19)	0.006	2.24 (1.36–3.69)	0.001
Preoperative Hemoglobin < 10 g/dL	1.54 (0.89–2.65)	0.11	-	
Combined surgery	0.61 (0.29–1.23)	0.16	-	
CPB time > 180 min	2.51 (1.19–5.25)	0.01	2.57 (1.19–5.34)	0.015
Cross clamping time > 60 min	1.23 (0.79–1.91)	0.35	-	
Administration of RBC products	1.56 (1.06–2.35)	0.03	1.41 (0.91–2.19)	0.12
Mechanical hemodynamic support	5.41 (1.09–27.9)	0.04	3.85 (0.69–21.2)	0.12

CABG: coronary artery bypass grafting; CPB: cardiopulmonary bypass; RBC: red blood cell; pMI: perioperative myocardial infarction. The multivariable model showed a good calibration as assessed by the Hosmer and Lemeshow goodness of fit test (*p* = 0.81) and fair discrimination as assessed by the receiver operating characteristic curve [area under the curve (AUC) 0.61; 95% CI 0.55–0.67, *p* = 0.0001].

## Data Availability

The dataset used and analyzed during the current study is available from the corresponding author upon reasonable request.

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
