# Peer review of "Five-Year Mortality of Patients with Perioperative Myocardial Infarction After On-Pump Isolated or Combined Coronary Artery Bypass Graft Surgery: A Retrospective Propensity Score-Weighted Analysis"

_jcm, 2025, doi:10.3390/jcm14227970_

Round 1
Reviewer 1 Report
Comments and Suggestions for Authors
This manuscript addresses an important and clinically relevant question: the long-term prognostic impact of periprocedural myocardial infarction (pMI), defined according to the Fourth Universal Definition, in patients undergoing isolated or combined on-pump CABG. The authors analyze a large single-center cohort using robust statistical techniques, including propensity-score weighting and landmark analysis, to explore five-year all-cause mortality.
The topic is timely and contributes valuable data to the ongoing discussion regarding pMI definition, incidence, and outcomes. However, there are some areas where methodological details, interpretation, and presentation could be strengthened to enhance the manuscript’s scientific rigor and clarity.
- Please use the term “periprocedural myocardial infarction (MI)” rather than “perioperative MI” to remain consistent with current literature and guideline terminology.
- In the Methods section, the authors state that patients undergoing elective or urgent on-pump CABG were included. However, when describing the diagnostic criteria for periprocedural MI, they report only that diagnosis required a cardiac troponin (cTn) elevation of more than ten times the 99th percentile. It should be noted that this threshold applies only to patients with normal baseline cTn levels (<99th percentile URL). For patients with elevated baseline values, documentation of a further rise of >20% from baseline is also required. Please clarify the exact diagnostic criteria applied in these cases.
- Also in the Methods section, please specify the type of troponin assay used — conventional vs. high-sensitivity, and cTnI vs. cTnT — and indicate whether sex-specific upper reference limits were considered.
- It would be useful to clarify how “urgent” versus “elective” cases were operationally defined.
- The mixture of isolated and combined CABG procedures introduces heterogeneity. Consider performing a sensitivity analysis stratified by surgical type (isolated vs. combined) to confirm that results are consistent.
- The authors conclude that pMI is mainly associated with early mortality but not with long-term risk among survivors. This is an important observation, but it should be tempered: the power for the landmark analysis appears limited given the small number of events after 30 days.
- Clinical Implications: The statement that “pMI does not compromise long-term outcomes for hospital survivors” might be overstated. Consider rephrasing to “no significant long-term difference was observed among survivors in this cohort.”
- It is suggested to emphasize more clearly in the Discussion that approximately one-third of the study population underwent CABG following an acute coronary syndrome (ACS). The finding that periprocedural MI had an adverse prognostic impact even in this subgroup should be highlighted, as it aligns with recent evidence in PCI patients (cite PMID: 39968630).
- The Discussion could be strengthened by integrating a brief section on practical clinical implications, for example the role of systematic troponin surveillance, postoperative imaging strategies, or structured smoking cessation programs, to better translate the study findings into actionable recommendations.
- Abstract: The sentence “This effect persisted after weighting (P = 0.041)” should include exact HR and CI for completeness.
Author Response
Reviewer 1:
This manuscript addresses an important and clinically relevant question: the long-term prognostic impact of periprocedural myocardial infarction (pMI), defined according to the Fourth Universal Definition, in patients undergoing isolated or combined on-pump CABG. The authors analyze a large single-center cohort using robust statistical techniques, including propensity-score weighting and landmark analysis, to explore five-year all-cause mortality.
The topic is timely and contributes valuable data to the ongoing discussion regarding pMI definition, incidence, and outcomes. However, there are some areas where methodological details, interpretation, and presentation could be strengthened to enhance the manuscript’s scientific rigor and clarity.
1.Please use the term “periprocedural myocardial infarction (MI)” rather than “perioperative MI” to remain consistent with current literature and guideline terminology.
Response: We thank the reviewer for this relevant remark. Both terms “periprocedural” and “perioperative” are used in the current literature. However, we decided to retain the term perioperative myocardial infarction, in accordance with the recent European consensus statement (EJCTS, 2024; https://doi.org/10.1093/ejcts/ezad415), as it more accurately reflects the surgical context of CABG, whereas “periprocedural” is generally associated with coronary angiography or PCI.
2.In the Methods section, the authors state that patients undergoing elective or urgent on-pump CABG were included. However, when describing the diagnostic criteria for periprocedural MI, they report only that diagnosis required a cardiac troponin (cTn) elevation of more than ten times the 99th percentile. It should be noted that this threshold applies only to patients with normal baseline cTn levels (<99th percentile URL). For patients with elevated baseline values, documentation of a further rise of >20% from baseline is also required. Please clarify the exact diagnostic criteria applied in these cases.
Response: We thank the reviewer for this insightful comment. The diagnostic criteria proposed in the Circulation consensus (4th Universal Definition of MI) apply primarily to periprocedural infarction during PCI, where baseline troponin values and their relative changes are measurable. However, our study focused on surgical patients undergoing CABG. In this context, postoperative myocardial infarction (pMI) was defined according to the European consensus for cardiac surgery, which relies on ECG and echocardiographic findings, since baseline cTn levels are often elevated in urgent cases. As clarified in the revised Methods section, for emergency coronary surgeries performed in the setting of acute coronary syndrome, diagnosis of pMI was based solely on new ischemic ECG changes and/or new regional wall motion abnormalities, in line with current recommendations [12].
We complete the method section, page 7 as follow: “For emergency coronary surgeries performed in the context of an ongoing acute coronary syndrome, postoperative cTn levels were not used for the diagnosis of perioperative myocardial infarction, as preoperative ischemia or infarction typically leads to pre-existing troponin elevation. In such cases, only new ischemic ECG changes and/or new regional wall motion abnormalities on echocardiography were considered for the diagnosis of postoperative myocardial infarction, in accordance with current recommendations [12].
- Also in the Methods section, please specify the type of troponin assay used — conventional vs. high-sensitivity, and cTnI vs. cTnT — and indicate whether sex-specific upper reference limits were considered.
Response: We thank the reviewer for this helpful comment. At our center, cardiac troponin I (cTnI) levels were measured using a conventional chemiluminescence immunoassay (ELISA sandwich method, Atellica® Immunoassay analyzer, Siemens). The 99th percentile upper reference limit was 0.045 µg/L, and sex-specific reference limits were not applied, as a single institutional cut-off value was used. This clarification has been added to the Methods section page 7 : “At our center, cTnI levels were measured using conventionnal chemiluminescence immunoassay (ELISA Sandwich on Atellica® Immunoassay analyzer, Siemens) at 0, 6, 24, and 48 hours after the primary cardiac surgery. The 99th percentile threshold was set at 0.045 g/L. No sex-specific reference limits were applied, as a single institutional cut-off value was used. ECGs were performed at 0, 6, 24, and 48 hours, and segmental kinetic abnormalities were assessed through TTE at 0, 24, and 48 hours.”
- It would be useful to clarify how “urgent” versus “elective” cases were operationally defined.
Response: We thank the reviewer for this relevant comment. In the revised manuscript, we have clarified that surgical urgency was defined according to international standards from the Society of Thoracic Surgeons (STS) and the ESC/EACTS guidelines (Neumann FJ et al., Eur Heart J 2019;40:87–165). Elective procedures referred to stable patients scheduled in advance without ongoing ischemia or hemodynamic compromise, whereas urgent surgeries were performed during the same hospitalization in patients with ongoing or recurrent ischemia, hemodynamic instability, or failed percutaneous or medical therapy requiring surgical revascularization within 24 hours. This clarification has been added to the Methods section page 6 as follow: “The urgency of surgery was defined according to international guidelines. Elective procedures referred to stable patients scheduled in advance without ongoing ischemia or hemodynamic compromise. Urgent surgeries were performed during the same hospitalization in patients with ongoing or recurrent ischemia, hemodynamic instability, or failed percutaneous or medical therapy requiring surgical revascularization within 24 hours [9]”
5.The mixture of isolated and combined CABG procedures introduces heterogeneity. Consider performing a sensitivity analysis stratified by surgical type (isolated vs. combined) to confirm that results are consistent.
Response: We thank the reviewer for this relevant comment. As shown in Table 2, there was no significant difference in the proportion of isolated versus combined procedures between the pMI and no-pMI groups (isolated CABG: 72% vs. 71%, P = 0.91; combined surgery: 28% vs. 29%, P = 0.82). Moreover, combined procedures were not associated with the occurrence of pMI in univariate or multivariable analyses (Table 3).
To further address this point, we specified in the Results section that, in the subgroup of patients undergoing isolated CABG (71%, n = 506/712), the 5-year all-cause mortality rate was higher in the pMI group (31%) than in the no-pMI group (12%), with a significant difference in survival curves (log-rank P < 0.001; Supplementary Figure 1).
Additionally, in the Limitations section, we emphasized that although our cohort included both isolated and combined CABG procedures, combined surgeries were not associated with either pMI or 5-year all-cause mortality, suggesting that this heterogeneity did not materially affect our findings; as follow: “In this study, we examined the impact of pMI in a mixed population undergoing both isolated CABG and combined cardiac procedures, which could potentially introduce bias. However, combined procedures were evenly distributed between groups and were not associated with either pMI or 5-year all-cause mortality, suggesting that surgical heterogeneity did not materially influence our findings”.
- The authors conclude that pMI is mainly associated with early mortality but not with long-term risk among survivors. This is an important observation, but it should be tempered: the power for the landmark analysis appears limited given the small number of events after 30 days.
Response: We thank the reviewer for this insightful comment. We fully agree that the limited number of late deaths after 30 days may reduce the statistical power of the landmark analysis and should be acknowledged. Accordingly, we have revised both the Limitations section and the Conclusion to temper our interpretation and better reflect the strength of our findings.
We change the limit section, page X, as follow : “In addition, although pMI was mainly associated with early mortality, the absence of a significant difference beyond 30 days should be interpreted with caution. The number of late events after 30 days was relatively small, which may have limited the statistical power of the landmark analysis and precluded detection of subtle differences in long-term mortality among survivors.”
And
Wer change the conclusion section page X as follow : Among patients surviving this critical period, long-term outcomes appear comparable, suggesting that the prognostic impact of pMI is largely confined to the immediate postoperative window
7.Clinical Implications: The statement that “pMI does not compromise long-term outcomes for hospital survivors” might be overstated. Consider rephrasing to “no significant long-term difference was observed among survivors in this cohort.”
Response: We thank the reviewer for this valuable suggestion. The sentence has been rephrased in the Discussion to better reflect the data and avoid overinterpretation; as follow: These findings emphasize the need to improve perioperative management to reduce the incidence of pMI, while indicating that, in this cohort, no significant long-term difference in survival was observed among hospital survivors.
8.It is suggested to emphasize more clearly in the Discussion that approximately one-third of the study population underwent CABG following an acute coronary syndrome (ACS). The finding that periprocedural MI had an adverse prognostic impact even in this subgroup should be highlighted, as it aligns with recent evidence in PCI patients (cite PMID: 39968630).
Response: We thank the reviewer for this valuable suggestion. We agree that the preoperative context of acute coronary syndrome (ACS), particularly NSTEMI, is clinically relevant as it may reflect increased myocardial vulnerability to ischemic injury. In our study, approximately one-third of patients underwent CABG following an ACS, predominantly NSTEMI. However, this variable was not independently associated with the occurrence of pMI in either univariate or multivariate analysis, likely due to the limited number of patients in this subgroup and the resulting lack of statistical power.
To acknowledge this point and align with recent findings showing the prognostic impact of periprocedural myocardial injury in NSTEMI patients undergoing PCI (Armillotta et al., Circulation 2025; 151:760–772, PMID 39968630), we have added a dedicated statement in the Limitations section as follows: Approximately one-third of patients underwent CABG in the context of an acute coronary syndrome, predominantly NSTEMI. Although this variable was not independently associated with pMI in our cohort, this may be partly explained by the limited number of patients and the resulting lack of statistical power. This subgroup likely reflects increased myocardial susceptibility to perioperative ischemic injury, consistent with findings in NSTEMI patients undergoing PCI, where periprocedural myocardial injury independently predicted adverse outcomes [31].
- The Discussion could be strengthened by integrating a brief section on practical clinical implications, for example the role of systematic troponin surveillance, postoperative imaging strategies, or structured smoking cessation programs, to better translate the study findings into actionable recommendations.
We thank the reviewer for this valuable suggestion. We have now added a dedicated paragraph in the Discussion under the heading “Clinical Implications”, highlighting actionable strategies derived from our findings. This new section emphasizes the preoperative identification of high-risk patients (e.g., recent ACS, diffuse coronary disease, prolonged clamp time), tailored surgical approaches, and standardized perioperative protocols integrating postoperative troponin surveillance and Δ-troponin thresholds to guide early coronary angiography or CT angiography when pMI is suspected. It also underlines the importance of structured long-term follow-up and secondary prevention programs. The following paragraph was added:
Clinical Implications
From a practical standpoint, our results emphasize the importance of preoperative identification of patients at high risk of pMI, such as those with recent ACS, diffuse coronary disease, or prolonged anticipated clamp times. In these patients, adopting tailored surgical strategies—including minimizing ischemic time or using less invasive or off-pump techniques when feasible—may help reduce myocardial injury. At the institutional level, standardized perioperative pathways integrating systematic postoperative troponin surveillance and predefined diagnostic thresholds (Δ-troponin) could allow early detection of pMI. In line with recent EACTS recommendations, this should prompt early coronary angiography or coronary CT angiography to confirm graft patency and guide reintervention when necessary. Such structured protocols could improve short-term outcomes by enabling prompt correction of reversible ischemia. In the longer term, implementing dedicated follow-up programs for patients with preoperative ACS or pMI—including multimodal imaging and aggressive secondary prevention—may mitigate the sustained risk of adverse events. These practical measures bridge the gap between research findings and real-world clinical application, providing a foundation for structured prevention and management of pMI after CABG.
10.Abstract: The sentence “This effect persisted after weighting (P = 0.041)” should include exact HR and CI for completeness.
Response: We thank the reviewer for this comment. The abstract has been updated to report the exact effect size for completeness, which is consistent with the main text derived from the IPTW-weighted Cox model: “This effect persisted after weighting (HR = 2.43, 95% CI [1.56–3.78], P = 0.041). The same result are reported in the Results section.

Reviewer 2 Report
Comments and Suggestions for Authors
Please see attachment.

Author Response
Reviewer 2
Comment 1: (1) Please clarity whether pMI, PMI, postoperative, perioperative refers to the same concept throughout the manuscript. If so, please use consistent terminology. If not, please clearly define and distinguish it within the scope of this study. Line 50: perioperative myocardial infarction (pMI) Line 186-187: pMI: postoperative myocardial infarction. Line 403: PMI: postoperative myocardial infarction.
Response: We thank the reviewer for this helpful comment. The terminology has been harmonized throughout the manuscript. The term perioperative myocardial infarction (pMI) is now used consistently, as it best reflects the definition according to the Fourth Universal Definition of Myocardial Infarction and the scope of our study.
(2) A similar issue is also observed for the variables CPB and CBP In Table 2, the variable is labeled as CPB time, but in the table 2 note it is written as CBP. Please verify Line 168: CPB time > 180 minutes… Line 247: cardiopulmonary bypass time > 180 minute Please clarity whether cardiopulmonary bypass time (CPB), coronary bypass (CBP), coronary artery bypass grafting refers to the same concept throughout the manuscript. Please clarity the difference between variables coronary bypass and coronary artery bypass grafting in this study. It is recommended to explain, either in literature review, study, or discussion section, the rationale for not combining these two variables into a single one.
(3) In line 247, the text describes “a cardiopulmonary bypass time > 180 min”, but in In Table 2, the variable is labeled as CBP time > 180 minutes, and in the table 2 note it is written as CBP: coronary bypass. Please verify.
Comment 2: Please check if there are any typos Line 73: -year all-cause mortality in a large retrospective cohort… It’s supposed to be five-year all-cause mortality Comment 3: Please check if there are any typos Line 183: (pMi group), and 600 (84%)… and Line 189: ... In the pMi group It’s supposed to be: pMI group
Comment 4: Line 189: Demographic and preoperative data were summarized in Table 1. Line 194: Table 1. Preoperative characteristics of the study population There is an inconsistency between line 189 and line 194 2
Response: All noted inconsistencies and typos have been corrected. The abbreviation CPB (cardiopulmonary bypass) is now used consistently throughout the manuscript, and CBP has been removed. The difference between CPB and CABG has been clarified. Minor textual corrections were also made (“five-year all-cause mortality,” “pMI group,)
Comment 5: It was mentioned on the line 134 that as appropriate, categorical variables were expressed as frequencies and percentages 134 and compared using the χ2 or Fisher exact test. In Table 1, the number and percentage of male gender were comparing using the χ2; the mean standardized difference is not applicable for categorical variable. It seems that mean standard differences are reported for categorical variable in Table 1. Is this because the study employed propensity score matching? In Table 1 and Table 2, other categorical variables also raise similar questions as the male gender variable require clarification.
Response: We thank the reviewer for this useful comment and for the opportunity to clarify. The mean standardized difference (SMD) was intentionally reported for all baseline variables, including categorical ones, as a measure of covariate balance between groups rather than a statistical test. The SMD quantifies the magnitude of imbalance independently of sample size and is widely used in studies employing propensity score–based analyses. In our study, the SMD was presented in Tables 1 and 2 to provide a consistent measure of baseline balance before weighting. For categorical variables, SMDs were computed using proportions, while for continuous variables, they were based on means. This approach was added to the table legends to avoid confusion.
Comment 6:
It was mentioned on the line 110-114 that preoperative data collected…active smoking…angina symptomatology classified according to the Canadian classification It was mentioned on the line 190-191 that and active smoking (n=83/112 vs. n=94/600, P=0.008) compared to … It appears that information on active smoking is missing from Table 1.
Response: We thank the reviewer for this careful observation. The terminology has been harmonized throughout the manuscript, and the variable has been consistently labeled as “active smoking” in both the text and tables. The information on active smoking was already included in Table 1 (line “Current smoker”) and has now been renamed “Active smoking” for consistency with the rest of the manuscript.
Comment 7: Could you please confirm whether angina symptoms and angina severity refer to two separate variables or the same variable in this study? If they are the same variable, please check whether the reported p value might be a typographical error. Line 190: more angina symptoms (n=57/112 vs. 304/600, P=0.04)… Page 6, table 1: ... Angina severity according to CCS > 2…P=0.05
Response: We thank the reviewer for this useful clarification. Angina symptoms and angina severity refer to the same variable, defined according to the Canadian Cardiovascular Society (CCS) classification. The text has been revised for clarity to specify “angina symptoms with a CCS class > II.” The corresponding p-value has been verified and corrected to P = 0.05 for consistency with Table 1.
Comment 8: We note that the p-value for Diabetes mellitus in Table 1 is reported as 0.008. Pleases confirm that this value is correct
Response: We thank the reviewer for this careful reading. The p-value of 0.008 in Table 1 corresponds to Active Smoking, not to Diabetes mellitus (for which the p-value is 0.91). We have double-checked the table to ensure accuracy, and the values are correct as presented.
Comment 9: It was mentioned on the line 135 that Continuous variables were expressed as mean± standard deviation (SD) and compared by unpaired Student t-test or the MannWhitney test. It was mentioned on the line 189-193 that Demographic … in Table 1.… the logistic EuroSCORE II was higher than in the no-pMI group (respectively, 6.5±4 % vs. 5.3 ±3 %, P=0.01) It seems that the EuroSCORE 2 variable in Table 1 was analyzed using a t-test rather 3 than logistic regress. Could you please confirm?
Response: We thank the reviewer for this helpful clarification. The term “logistic” referred to the official name of the EuroSCORE II model and not to a regression analysis. To avoid any confusion, we have removed the word “logistic” throughout the manuscript. The comparison of EuroSCORE II values between groups was performed using an unpaired Student t-test, as described in the Methods section.
Comment 10: Please check if there are any typos Line 206: Administration of vasopressor is n=103/600, 21%; P=0.003… It appears that the administration of vasopressor variable in the No pMI is reported as 123
Comment 11: Please check if there are any typos Line 208: rates of AKI (n=26/112, 32% vs. n=97/600, 16%; P=0.0001) … It appears that the acute kidney injury variable in the pMI is reported as 36(32) and P
Comment 12: Please check if there are any typos (Terminology should be consistent) Line 219: After IPW adjustment, the me… Is it supposed to be IPTW ?.
Response: We thank the reviewer for these careful observations. All typographical and consistency issues have been verified and corrected in the revised version. The numbers for vasopressor use and acute kidney injury (AKI) have been checked against the original dataset and are now reported correctly in both the text and tables. In addition, the terminology has been standardized, and “IPW” has been replaced with “IPTW” throughout the manuscript for consistency.

Round 2
Reviewer 1 Report
Comments and Suggestions for Authors
Thank you to the authors for the revisions made, which I believe have enhanced the quality of the final manuscript. I have no further comments.